# The smashHitCore Ontology for GDPR-Compliant Sensor Data Sharing in Smart Cities

**DOI:** 10.3390/s23136188

**Published:** 2023-07-06

**Authors:** Anelia Kurteva, Tek Raj Chhetri, Amar Tauqeer, Rainer Hilscher, Anna Fensel, Kevin Nagorny, Ana Correia, Albert Zilverberg, Stefan Schestakov, Thorben Funke, Elena Demidova

**Affiliations:** 1Semantic Technology Institute (STI), Department of Computer Science, Universität Innsbruck, 6020 Innsbruck, Austria; a.kurteva@tudelft.nl (A.K.); Tek-Raj.Chhetri@uibk.ac.at (T.R.C.); or amar.tauqeer@sti2.at (A.T.); rhilscher@rti.org (R.H.); 2Industrial Design Engineering, Delft University of Technology, 2628 CE Delft, The Netherlands; 3Consumption and Healthy Lifestyles Chair Group, Wageningen University & Research, 6706 KN Wageningen, The Netherlands; 4RTI International, Research Triangle Park, NC 27709, USA; 5Wageningen Data Competence Center, Wageningen University & Research, 6708 PB Wageningen, The Netherlands; 6Institut für Angewandte Systemtechnik Bremen GmbH (ATB), 28359 Bremen, Germany; nagorny@atb-bremen.de (K.N.); correia@atb-bremen.de (A.C.); zilverberg@atb-bremen.de (A.Z.); 7L3S Research Center, Leibniz University Hannover, 30167 Hannover, Germany; schestakov@L3S.de (S.S.); tfunke@L3S.de (T.F.); 8Data Science and Intelligent Systems Group (DSIS), University of Bonn, 53115 Bonn, Germany; elena.demidova@cs.uni-bonn.de

**Keywords:** ontology, consent, contracts, sensors, data sharing, GDPR, smart cities, insurance

## Abstract

The adoption of the General Data Protection Regulation (GDPR) has resulted in a significant shift in how the data of European Union citizens is handled. A variety of data sharing challenges in scenarios such as smart cities have arisen, especially when attempting to semantically represent GDPR legal bases, such as consent, contracts and the data types and specific sources related to them. Most of the existing ontologies that model GDPR focus mainly on consent. In order to represent other GDPR bases, such as contracts, multiple ontologies need to be simultaneously reused and combined, which can result in inconsistent and conflicting knowledge representation. To address this challenge, we present the smashHitCore ontology. smashHitCore provides a unified and coherent model for both consent and contracts, as well as the sensor data and data processing associated with them. The ontology was developed in response to real-world sensor data sharing use cases in the insurance and smart city domains. The ontology has been successfully utilised to enable GDPR-complaint data sharing in a connected car for insurance use cases and in a city feedback system as part of a smart city use case.

## 1. Introduction

The GDPR [1], which came into effect in 2018, has introduced six legal bases, namely consent, contract, legal obligation, vital interest, public task and legitimate interests for the lawful processing of personal data (see [2]). Consent is arguably the most widely used GDPR basis and specific requirements apply to it. It should be specific, unambiguous, freely given, and informed and should be implemented in a way that it can be withdrawn as easily as it was given (Art. 7, Rec. 32, 42, 43). However, depending on the type of data processing and context a contract might be needed instead [2]. In comparison to consent, contracts have more restrictions in the form of terms and conditions with specific obligations (Art. 6(1)(b)), which depend on the nature of the contract (e.g., sale of goods, e-commerce, education, services) [3].

In smart cities one’s data is spread across different sources (such as vehicles sensors, smart home sensors, databases, people, etc.) and can be used for different purposes by different entities simultaneously. For example, an individual can report a damaged road with a photo and location information (e.g., GPS data) to authorities so that authorities can take appropriate action. The main challenge is the management of the consent itself as each entity (e.g., company) might store it in different formats and locations. Locating and analysing all consent information is time consuming and still a challenge for many small and medium enterprises (SMEs) as discussed in [4]. Insurance companies handle personal data on a daily basis, making them one of the industries highly impacted by the GDPR. One of the main challenges when modelling consent for insurance companies is the existence of dynamic situations such as a change in ownership of the insured property (e.g., a car), which can disrupt existing declarations of consent. In our study, we have considered two sensor data sharing use cases, where GDPR compliance is expected. Use case one (UC1) focuses on vehicle sensor data sharing in smart cities, while use case two (UC2) focuses on vehicle sensor data sharing in the insurance domain. Details about each use case are presented in more detail in Section 2.

Semantic Web technologies, such as ontologies, can represent diverse dynamic contexts and support faster and easier knowledge discovery, data interoperability, and transparency [3,5,6]. Further, ontologies help translate human knowledge into a machine-understandable format to be used as a knowledge management infrastructure within computer systems [7]. These capabilities of ontologies make them a well-fitting solution for the representation and interpretation of the heterogeneous data in UC1 and UC2. As presented later in Section 3, several semantic models for consent, contracts and sensors have already been built. However, our analysis in [8] and the requirements of UC1 and UC2 point to three areas of improvements that require an addition to the set of existing ontologies. First, the concepts of consent and contract are treated as exclusive with ontologies modelling either consent or contract. Second, specific data sources such as sensor data are in most cases not modelled in detail (e.g., what specific sensor is the source of the data). Furthermore, finally, not all ontologies are openly available. Consent for data processing should be requested in an informed manner. Individuals need to be presented with information about the type of data that will be processed and the specific purpose of the processing (e.g., GDPR Art. 4, 6, 7 and Rec. 32, 42). For example, consent for vehicle sensor data sharing is an ambiguous request as a vehicle has multiple sensors for different observations. To be compliant with the GDPR, one should know exactly the type of sensor and the corresponding data the consent covers (e.g., GPS sensor data and more specifically latitude and longitude data). Third, existing ontologies are limited when it comes to modelling specific concepts needed for complex scenarios such as broken consent chains (e.g., due to the transfer of property ownership [9]).

In our UC1 and UC2 it is also possible that a contract is the adopted GDPR legal basis for data sharing and processing. This can be due to (i) data subjects being presented with a contract instead of a consent request (depending on the company) and (ii) data subjects interested in negotiating the terms and conditions for the data processing. When dealing with consent the obligations for the data subject are pre-defined, while contracts allow their negotiation. In UC1 and UC2, we have considered the possibility of both business to customer (B2C) and business to business (B2B) contracts, where the involved contractors can negotiate specific terms and conditions for data processing. Data selling between companies is a typical scenario where a contract is required (and consent is not enough). Another difference is that individuals have the right to revoke their consent at anytime, while contracts have specific steps that need to be followed for cases such as contract termination. To summarise, the use cases that motivated our work consider two GDPR legal bases for (sensor) data sharing—consent and contracts. To our knowledge, an ontology that consistently represents both concepts, dependencies between them and the relevant sensor data, has not yet been built.

Based on the need for such an ontology highlighted by smashHit’s [10] UC1 and UC2 and the GDPR (Art.6, Rec. 32, Rec. 44), we present the smashHitCore Web Ontology Language (OWL) [11] ontology. The ontology and its accompanying documentation are publicly accessible at [12,13].

smashHitCore, which is openly available, reuses and extends existing ontologies to provide a harmonised model for consent, contracts, sensor data and its processing. smashHitCore can be used in simpler cases such as informed consent modelling based on GDPR or more complex ones such as automatic contracting (i.e., automatic contract generation, signing and execution) and automated compliance verification based on one’s informed consent (see [9,14]).

The rest of the paper is structured as follows. The use cases that motivated our work are presented in Section 2. Section 3 presents an overview of existing ontologies for consent, contracts and sensor data, while Section 4 presents the methodology we followed for the development of smashHitCore. The ontology specification is presented in Section 5. The specific applications of smashHitCore for consent and contract compliance is presented in Section 6. Section 7 presents the evaluation of smashHitCore. Conclusions can be found in Section 8.

## 2. Use Cases

The smashHitCore ontology is based on two sensor data sharing use cases. UC1-Smart City Services is presented in Section 2.1, while UC2-Insurance Services is presented in Section 2.2.

### 2.1. UC1-Smart City Services

The increasing volume of smart city sensor data about traffic, road conditions, carbon emissions and the vehicles contributing to all these, holds great potential to be utilised for optimising current mobility services and for implementing more efficient ones in the future [15]. Helsinki is an example of a leading smart city, which has set as its goal to become one of the most functional cities in the world. For Helsinki, functionality means above all ensuring convenient everyday life for its citizens, visitors, and businesses by fully harnessing the potential of digitization. The aim of the city is to enhance the utilization of data and analytics, to produce individually tailored services for the city residents, proactively and when they need them. The city is committed to the MyData [16] principle, according to which residents must be able to manage the personal data that the city collects (e.g., authorise its use for different purposes and services, be able to restrict and even deny access to it). For example, personal data (e.g., vehicle sensor data such as fuel, speed, location) is essential for the current traffic planning and management digital services. Fluency and safety in city traffic have an enormous influence on the well-being of the residents, visitors, and businesses. Optimizing traffic flows and enhancing safety can only be conducted through a joint effort of the city, its residents, visitors, and businesses. However, such vehicle sensor data are considered personal, which means it should be shared and processed in a GDPR-compliant way that does not endanger one’s privacy. To be GDPR-compliant, informed consent needs to be requested, received and managed through its whole lifecycle [8]. The main challenge is to perform all of the above for the heterogeneous sensor data at scale while complying with the GDPR.

### 2.2. UC2-Insurance Services

UC2 focuses on insurance services that also rely on vehicle sensor data sharing [17]. With the growing ability of vehicles to generate, gather and share car data with third parties among different data-sharing platforms, there is a need for flexible and easily manageable procedures to handle data owner consent and legal compliance, to achieve effective and traceable contracting. Current fleets of connected vehicles cannot provide the data required to support advanced usage-based insurance models without additional intervention during the workflow between vehicles and insurers. The challenges and complexities of the GDPR directives make cumbersome mechanisms necessary to gain, record, and manage consent and contract. The concerns of personal data misuse are also rising. The combination of understanding and relating to the value proposition, individuals’ trust, and cumbersome consent and contract processes result in a low opt-in rate for connected vehicle data exchange and particularly for connected insurance programs. Another concern is the possibility of sensitive data (e.g., GPS location, driving trends) leakage, which can be a major threat to both Original Equipment Manufacturers (OEMs) and data processors alike. Preserving individuals’ privacy and using their data in a GDPR-compliant way is a necessity.

## 3. Related Work

This section presents an overview of existing ontologies that model consent as defined by GDPR, contracts and the data related to them, specifically sensor data.

### 3.1. Semantic Models for Consent

This section is based on our previously conducted survey [8] on GDPR consent ontologies. One of the earliest ontologies that semantically represents consent as defined by the GDPR is the Consent and Data Management Model (CDMM) by Fatema et al. [18]. CDMM, available since 2017, represents consent as an entity within the context of a privacy policy. The ontology is able to represent consent through its whole life-cycle (i.e., from its request to its withdrawal). However, CDMM does not model meta-information such as data storage location, contact information of entities such as data controller, third parties and their obligations, which can be helpful when performing GDPR compliance verification.

Pandit et al. [19] present a GDPR-compliant consent ontology called GConsent [20]. The ontology’s main focus is to model consent-related information as defined by GDPR [19]. Consent is semantically represented as an artefact, that has multiple states such as given, refused, revoked and unknown. In comparison to CDMM, GConsent focuses on GDPR compliance and models consent in more detail. However, it does not model specific responsibilities required for our use cases such as the entity responsible for consent, specific contact information and obligations regarding the use and revocation of consent.

The PrOnto ontology [21] focuses on modelling GDPR obligations and requirements such as consent for personal data processing. Further, PrOnto models the specific documents, data, actors involved in different data processing and the legal rules that can apply. However, PrOnto is not openly available and its proprietary status prevents a wider adoption by researchers and industry practitioners.

The Legal Compliant Ontology to Preserve Privacy for the Internet of Things (LloPY) [22] is an ontology that models GDPR concepts, such as consent, but is not limited to it. The ontology further reuses privacy definitions provided by the National Institute of Standards and Technology (NIST) [23]. LloPY focuses on GDPR-compliant consent modelling for cases such as sensor data sharing. It models consent decisions, retention, disclosure, as well as types of sensor data by reusing the Semantic Sensor Network ontology (SSN) [24]. Despite its potential to model diverse data, LloPY is not openly available, which is an obstacle to its wider evaluation and reuse by the legal and Semantic Web communities.

The Business Process Re-engineering and Functional Toolkit for GDPR Compliance (BPR4GDPR) [25] represents consent, its states (e.g., provided, refused, revoked) and GDPR-related information that is needed for performing compliance checking. Based on its specification in [25], BPR4GDPR models specific roles, event types, context types and state types related to consent. However, BPR4GDPR is not open-access.

Consent is modelled as a policy and is used for privacy compliance checking by both SPECIAL’s Usage Policy Language (SPL) [26] and SPECIAL Policy Log Vocabulary (SPLog), which were built for the SPECIAL [27] project. The focus of both SPL and SPLog is to represent data usage policies in a machine-readable format and to define permissions based on one’s consent decision [26].

The Data Privacy Vocabulary (DPV) [28] models consent and its attributes (e.g., expiry time) in the context of privacy policies, which can be used for compliance checking. DPV models other GDPR-related concepts such as notice, expiry date and provision. The ontology has been extended with specific knowledge about GDPR’s legal basis, technical and organisational measures and processing categories (also known as DPV-GDPR [29]. A current limitation, with regards to GDPR and consent, is that DPV does not model the responsibilities of data controllers [8].

### 3.2. Semantic Models for Sensor Data

Russomanno et al. [30] present an approach for constructing a sensor ontology that is a formal conceptualisation of sensors. The presented OntoSensor [30] ontology is based on the Sensor Model Language (SensorML) [31] and the Suggested Upper Merged Ontology (SUMO) [32] and models sensors’ attributes, performance, capabilities and reliability. However, it has a limited expressivity with regard to describing the sensor’s observations, which are of significant importance for our UC1 and UC2.

Another widely used sensor ontology is SSN [24], a standard W3C recommendation in the field. SSN describes sensors, their observations, the involved procedures, the studied features of interest, the samples used to do so, the feature’s properties being observed or sampled, as well as actuators and the activities they trigger [33]. The ontology has been reduced by horizontal and vertical modularisation, which allows different users to only use domains that they are interested in.

The Sensor, Observation, Sample, and Actuator (SOSA) [34] ontology, which is the lightweight version of SSN, is event-centric. It focuses on modelling observations, sampling, actuations and procedures of sensors. While focusing on the interactions between sensors, and being easy to understand by a broader audience, the ontology does not provide the needed level of granularity of the sensors’ themselves (e.g., specific types of sensors such as GPS). However, SOSA can still be reused for cases where sensor observations are the main focus.

The Ontonym-Sensor [35] ontology provides a high-level description of sensors and capabilities such as their frequency, coverage, accuracy, and precision pairs. In addition, sensor observations (observation-specific information, metadata, sensor, timestamp, and the time period over which the value is valid, the rate of change) are modelled as well [35]. Due to its generic nature, Ontonym-Sensor can be reused in different scenarios that involve both sensors and sensor data.

Last but not least, Eid et al. [36] proposed the Sensor Data Ontology (SDO), which reuses the Suggested Upper Merged Ontology (SUMO) [32] for general definitions and associations and follows IEEE 1451.4 [37] (a standard for describing smart transducers and their observations). Although Eid et al. present an ontology, the main focus of their work in [36] is on sensor data search optimisation. Due to the limited information available, SDO can be viewed as lightweight foundational semantic model for sensors that can be built upon.

### 3.3. Semantic Models for Contracts

The Multi-Tier Contract Ontology (MTCO) by Kabilan and Johannesson [38] comprises three layers. The authors defined the conceptual models of contracts in the first layer. The specific contract types are defined in the second layer. The third layer is responsible for capturing contractual obligations and their fulfilment patterns. Furthermore, the proposed ontology contains different stages, such as drafting, negotiation, and signing, which can be useful in modelling semantic contracts. The performance obligations, rules, rights, and payments are additional contract details added by MTCO. The ontology does not, however, clearly distinguish between an electronic contract and a conventional contract.

In [39], the authors propose a contract ontology, which defines three building blocks, such as agreements amongst persons, promises, and considerations to support modelling semantic contracts. The authors describe different types of contracts (e.g., verbal and written), and events related to the execution, fulfilment, and exchange of contracts. The formalisation of the ontology in OWL and the modelling of specific contract domains (e.g., sales), however, are left for future work. The ontology also predates GDPR, therefore its particular legal requirements for data processing have not been taken into account.

The Financial Industry Business Ontology (FIBO) [40,41,42], which is a collection of eleven distinct ontologies that define entities and processes in the business and finance domains, is a more recent ontology for contracts, while FIBO does not concentrate on particular laws such as the GDPR, it does offer a thorough semantic model of concepts like contracts and agreements that can serve as the basis for any ontology focusing on GDPR. Although FIBO does not explicitly consider GDPR when modelling contracts, there have been recent updates made regarding how the law is mapped out.

### 3.4. Summary

Based on our previous ontology survey in [8] and the related work overviewed in Section 3, we have identified several ontologies that can represent consent as a GDPR basis for lawful data processing. GConsent focuses specifically on representing informed consent as defined by GDPR but does not model specific sensors and sensor data, which are needed in cases such as UC1 and UC2 (see Section 1). CDMM, SPL and DPV focus on privacy policies. PrOnto [21] and Llopy [22] model GDPR-compliant consent, however, both are not open access, which limits their reuse. Further, the ontologies that model GDPR focus exclusively on the concept of consent. Contracts, which are also a GDPR basis for data processing, are rarely modelled by consent ontologies. Similarly, the ontologies that model contracts (e.g., FIBO), do not model consent and do not follow a specific law such as GDPR. Sensors are rarely represented in consent and contract ontologies as a type of data source. The wide availability of ontologies for consent and contracts makes their reuse complex and time-consuming as it is not clear, which ontology to reuse, when, and how exactly.

## 4. Methodology

The smashHitCore ontology is a result of a collaborative effort of individuals from various backgrounds (i.e., Semantic Web, Privacy and Security, Law, Insurance and Mobility). We followed the methodology for ontology development by Noy and McGuiness [43] and used the VocBench [44] environment for collaborative ontology development. The development of smashHitCore [12] follows a hybrid approach, which is a combination of the ontology modelling guidelines presented by Uschold et al. [45] and Noy [43]. The following steps were carried out to build the ontology:Use case identification and analysis of the range of intended users and specific use-case related requirements (i.e., competency questions) (see Table 1) with industry collaborators.Identification of the main concepts needed for consent and contracts in UC1 and UC2. Review of GDPR requirements for informed consent with legal experts.Gathering and analysis of existing ontologies for consent (see [8]), contracts, sensor data and data processing.Creation of the smashHitCore ontology by reusing specific classes and their respective properties from existing ontologies. Definition of new use case-specific concepts.Preservation of consistency by following the “isA” relationship between subclasses and classes (e.g., Sensor isA Device). Capitalisation of each word when labelling classes (e.g., Data Source) and following the British English language standard.Evaluation of the expressivity of the ontology with the competency questions derived during Step 1 (see Table 1).Evaluation of the correctness of the ontology with the HermiT [46] reasoner.Reviewing and editing of the ontology again if needed.

## 5. The smashHitCore Ontology

This section presents an overview of smashHitCore’s main concepts (i.e., consent, contracts and licenses, information entity, agent, role, data source, resource, data processing, location and time) and how they relate to each other. smashHitCore consists of 202 classes, 87 object properties, 42 data properties and reuses 9 ontologies: GConsent [19], DPV [28], FIBO [41], PROV-O [47], OntoSensor [30], schema.org [48], Data Catalog Vocabulary (DCAT) [49], CampaNeo [50] and Languages, Countries, and Codes (LCC) [51]. The description of the ontology follows the guidelines for minimum information for the reporting of an ontology (MIRO) [52]. The full specification of smashHitCore is available online [13].

### 5.1. Consent

To model the concept of consent in a GDPR-compliant manner based on Art 4(11), smashHitCore reuses the class *gconsent:Consent* from GConsent [19]. Consent is a dynamic entity (i.e., its status and purpose can change over time) that can have status (*smashHitCore:ConsentStatus*) such as *gconsent:Invalid, gconsent:Withdrawn, gconsent:Valid, smashHitCore:Granted*. Further, consent has provenance information such as *smashHitCore:GrantedAtTime, smashHitCore:WithdrawnAtTime, smashHitCore:atLocation* that is associated with it.

To model detailed information about the context in which a consent decision is made, we have reused the *gconsent:Medium* class, which describes the medium through which consent was provided (e.g., web form, app, paper). Consent can be associated with a specific purpose via the *dpv:hasPurpose* object property with range the class *dpv:Purpose* (e.g., *dpv:CommercialInterest*, *dpv:Security* etc.). Consent can be related to specific data source and processing via the properties *smashHitCore:isAboutData* and *smashHitCore:forDataProcessing*. Any type of processing *smashHitCore:hasInput* and *smashHitCore:hasOutput* (e.g., sensor data from a specific sensor such as *sensor:GPS*). An overview of the class consent (in white) and relationships to other classes (in blue) are presented in Figure 1.

### 5.2. Contracts

smashHitCore reuses *fibo-fnd-agr-ctr:Contract* (Figure 2) to represent contracts and contractual obligations related to data sharing for UC1 and UC2. A challenge here is to provide all the necessary information needed for a contract while having a generic enough semantic model that can be utilised in diverse contract-based scenarios. The class *fibo-fnd-agr-ctr:MutualContractualAgreement* from FIBO [41], which can model contracts for both UC1 and UC2 was reused in smashHitCore. We have represented two widely used types of contracts, namely *smashHitCore:BusinessToBusiness* and *smashHitCore:BusinessToConsumer*. The object property *smashHitCore:hasDataProcessor* links a contract to a specific contract agent such as *smashHitCore:DataProcessor*.

Contracts are comprised of various building blocks such as terms and conditions, which can be linked to a contract via the object property *samshHitCore:hasTermsAndConditions*. When a contract has status *smashHitCore:Created*, the obligations associated with it become active and (i.e., all contractors need to start adhering to them). If a contract has expired, then all obligations become invalid. To capture this information, we have modelled different obligation states with the class *smashHitCore:ObligationState*, namely *samshHitCore:Invalid*, *samshHitCore:Valid*, *samshHitCore:Pending*, *smashHitCore:Fulfilled*.

Metadata such as the contract’s creation date and its effective date can be recorded as well with the object properties *smashHitCore:hasCreationDate* and *fibo-fnd-agr-ctr:hasEffectiveDate*. The object property *smashHitCore:hasExpirationDate* has been defined to represent the expiration date of a contract, while *smashHitCore:hasEndDate* refers to the date when a contract is terminated ahead of its expiry date. To ensure the integrity of contracts and to allow contract and identity verification, the data property *samshHitCore:hasSignature* can be used to store contractors’ signatures (in xds:string format).

### 5.3. Information Entity

The class *smashHitCore:Information Entity* (Figure 3) models concepts that are used to provide specific details about consent and contracts. The subclass *smashHitCore:SensorData Category* describes several categories of sensor data needed for our UC1, specifically *smashHitCore:BuildingData, smashHitCore:CarData* and *smashHitCore:RoadData*.

Four types of personal data (*dpv:PersonalDataCategory*) have been modelled based on GDPR’s Art. 4.1 (i.e., *dpv:Internal*, *dpv:External* and *dpv:Tracking*). These types of data are needed for processes such as user verification within the smashHit project, for insurance purposes based on UC2 and for modelling contracts. The terms and conditions (*smashHitCore:TermsAndConditions*) as well as the policy (*odrl:Policy*) are an essential part of any contract and license. The terms and conditions are a set of special conditions for a contract, while a policy defines a set of prohibitions and permissions [53].

### 5.4. Agent and Role

The class *prov:Agent* (Figure 4) represents three types of entities (*prov:SoftwareAgent, prov:Person, prov:Organization*) that are currently present in UC1 and UC2. Several types of organisations were reused from the FIBO [41] and CampaNeo [50] ontologies (*fibo-fbc-fe-fse:InsuranceCompany, campaneo:AutomobileOrganisation*, etc.). When designing the ontology we also considered that the role (class *dcat:Role*) of an agent can change over time and that an agent can have multiple roles depending on a given context and as defined by GDPR (*smashHitCore:DataController, smashHitCore:DataSubject, smashHitCore:DataProcessor, smashHitCore:DataProtectionOfficer*). An agent can be linked to a current or past role by using the *gconsent:hasRole* and *dcat:hadRole* object properties. Further, agents have personal data associated with them, which is represented by the class *dpv:PersonalDataCategory*. In cases such as GDPR compliance verification when all entities need to be notified if non-compliance has been detected, smashHitCore models a contact point (*dpv:Contact*) of an agent with the classes *dpv:Email* and *dpv:TelephoneNumber*.

### 5.5. Data Source and Resource

The class *dpv:DataSource* models the source of the data for which consent, contract or license is needed. The source can be a device such as a sensor (see Figure 3) (the class *sensor:Sensor* is reused from OntoSensor [30]), which models a wide spectrum of sensor concepts relevant to our UC1 and UC2. GDPR requires consent to be specific and unambiguous thus we have also reused specific types of sensors such as *sensor:GPS*, *sensor:Motion*. We have also defined complex sensors such as *smashHitCore:PhotoelectricSensor, smashHitCore:Magnetometer, smashHitCore:AirPollutionSensor*. Further, the concept of a resource (*dcat:Resource*) is reused from DCAT [49]. A resource is an asset such as a *dcat:Dataset, smashHitCore:PersonalData* or *smashHitCore:SensorData*, which has been produced by an agent. For example, in UC1 and UC2, a data set generated with vehicle sensor data can be both the input and the output of data processing.

### 5.6. Data Processing

Consent must be given for a specific type of data processing. smashHitCore reuses the concept of data processing from DPV [28] and relevant concepts that describe different types of data processing such as *dpv:Adapt*, *dpv:Align*, *dpv:Collect*, *dpv:Record* and *dpv:Use*. We have limited the concepts to those events directly listed within GDPR (Art. 4 (2)) and hence the result is similar to the processing concepts from GConsent [19]. Each data processing event has input and output, which we represent via the *rdf:hasInputData* and *rdf:hasOutputData* properties.

### 5.7. Location and Time

To represent the location and time of consent and contract status updates (e.g., when and where consent was granted or a contract was signed), smashHitCore reuses the concepts *LCC:Location* and *time:TemporalEntity*. The *LCC:Location* class has the subclass *LCC:GeographicRegion*, which is used to describe the geographic location where the status of consent or a contract changes. Its subclass *smashHitCore:StorageLocation* represents the technical storage location defined by a URL (e.g., a URL of a server where the data is stored).

The class *time:TemporalEntity*, which is reused from OWL-Time [54], is a temporal sequence that represents a time instant (e.g., the expiration date of consent or a contract) or a period of time (e.g., duration of data progressing, consent or a contract). *time:TemporalEntity* has two subclasses *time:Interval* and *time:Instant*. The object properties *time:hasBeginning* and *time:hasEnd* can be used to express the start and end of a time instant.

## 6. Application of smashHitCore for GDPR Compliance

smashHitCore, which is a basis for the smashHit legal knowledge graph (KG) (see Figure 5b in [9]), has been used in the compliance verification tool [9,14] that is developed as a part of the smashHit project to enable legally compliant data exchange in the smart city and insurance use case scenarios (see Section 2). Our work [9,14], which makes use of the smashHitCore ontology, has been evaluated in real-world insurance and smart city scenarios. Details regarding the integration and testing are available in [55,56].

The semantic annotation of one’s informed consent and contracts (i.e., building the KG), their management and use for compliance are the main tasks of our automated compliance verification tools presented in [9,14]. In [9,14], by following a microservices [57] architecture, which supports scalability (see [58]), we have presented two tools that fully automate the GDPR compliance verification processes for both consent and contract. The tools adhere to the principle of data protection by design (Rec. 78) and are built around UC1 and UC2 modelled by the smashHitCore ontology. Moreover, Knoke and Iheanyi [59] provide details on the technical and organisational measures that our work implements for data protection by design principles from the maturity model perspective.

The consent compliance verification tool [9], which makes use of the smashHitCore ontology, is comprised of several components, each having a different task. Data and consent management are performed by the data processing module, which also deals with the execution of different SPARQL [60] queries (i.e., consent annotation, consent revocation) based on the smashHitCore ontology. The consent module in the tool performs the validation of the input consent, which is received in JavaScript Object Notation (JSON) and also transforms it to the KG following the smashHitCore ontology. A hybrid algorithm, combining both asymmetric and symmetric encryption, is used to ensure the integrity of the information and to make the KG access secure. Due to the sensitive information stored in the KG, it is not openly available. Its use within smashHit is perfomed via secured APIs (see Figure 6 in [9]). Details on the automated compliance verification tool, its mechanism and how compliance is performed with the help of smashHitCore are presented in [9]. In addition, smashHitCore is used as the main schema for building GDPR-compliant consent request forms online as showcased in [61] and for visualising post-consent data flows with KGs in [62].

## 7. Evaluation

In this section, we present the evaluation of smashHitCore in terms of ontology engineering, expressivity (ability to represent diverse GDPR and sensor data concepts) and its application for automated consent and contract compliance. Section 7.1 provides information on smashHitCore’s evaluation with a set of competency questions and standard ontology evaluation tools, while Section 7.2 provides details on the performance analysis of the tools for consent and contract verification that utilise smashHitCore.

### 7.1. Ontology Evaluation

To evaluate smashHitCore’s granularity and completeness with regards to UC1 and UC2, a set of competency questions (see Table 1) for consent, contracts, data processing, and corresponding agents was used. The competency questions for consent presented in Table 1 were derived from GDPR’s requirements for informed consent (Art. 7, Rec. 32, 42, 43) and are similar to the ones used in [19]. We have also derived similar competency questions for contracts which are based on GDPR’s requirements for data-sharing contracts (Art. 25 and 32). For each competency question we provide the set of relevant concepts and object properties that can be used to model the answer. Similarly to [19], smashHitCore was also evaluated with the HermiT [46] and Pellet [63] reasoners, and no inconsistencies were found. The OOPS! [64] ontology pitfall scanner was used as well to detect and correct issues in our semantic model.

The evaluation with the competency question in Table 1 shows that our ontology can model informed consent based on GDPR’s requirements and can further provide details about specific types of sensors and sensor data, which is not possible when using ontologies such as GConsent [19], CDMM [18] in a stand-alone way. smashHitCore represents different types of agents, contact information and specific types of personal data (e.g., email, username, address), which are not represented in detail in ontologies such as CDMM, GConsent, SPL and SPLog [26]. Such information can be useful for processes such as consent verification and compliance checking as presented in [4,65]. Further, smashHitCore also models contracts, which allows it to be reused in scenarios where consent is not enough for the legal processing of data. In such cases, existing contract ontologies such as FIBO [41] can be reused but they rarely model consent. smashHitCore provides a semantic representation of these concepts in one place.

### 7.2. Interoperability and Performance Evaluation

The two main tools presented in [9,14], which utilise the smashHitCore ontology, were developed as a part of the smashHit project. These tools were further used by other use case-specific software components that were also part of the smashHit project, namely data use traceability [66] and the context sensitivity solution [67] in the connected car and smart city feedback application [55]. Therefore, we evaluate the interoperability and performance based on the application of the smashHitCore ontology.

**Interoperability.** The interoperability of the smashHitCore ontology is evaluated using three dimensions: (i) semantic interoperability, (ii) technical interoperability, and (iii) organisational interoperability, following Ducq and Chen [68] and Guédria et al. [69]. Semantic interoperability enables a system to combine information from heterogeneous sources and process it in a meaningful manner, and organisational interoperability focuses on business goals, i.e., use cases in our case. Similarly, organisational interoperability is concerned with interconnecting different services including data integration and exchange [69]. The smashHitCore ontology models all of the necessary information needed for representing consent and contract, that is required by the use cases in order to share data and design applications, such as user interfaces, in a standardised manner [70], thereby achieving organisational interoperability. As with organisational interoperability, the smashHitCore ontology also meets the requirement of semantic interoperability, as it enables the exchange of the same meaning of information across different systems in a machine-readable format. For instance, all software components, such as data use traceability [66], can interpret the same meaning as [9] for consent-related information, meeting the requirement of semantic interoperability. The third dimension is technical interoperability. For technical interoperability, we check to see if the ontology in any way supports technical interoperability, enabling data exchange and the integration of various software components. In our case, the smashHitCore ontology has supported the development of an application based on a common specification and has enabled the seamless integration of multiple software components, including those from use cases [55,66,70]. We can therefore conclude that the smashHitCore ontology facilitates interoperability. Similar to interoperability, the use of the smashHitCore ontology in the applications did not impede scalability and had an acceptable performance for our use cases [9,14].

**Performance.** The performance of the smashHitCore ontology is evaluated by measuring the execution/processing time of queries to/from GraphDB [71]. This is performed for both of the applications of smashHitCore (see Section 6) based on the use cases discussed in Section 2. As an example, in the Contract Compliance Verification (CCV) [14] tool, a contract creation process (i.e., annotating standard contract information such as terms and conditions, clauses, contracting party information, and party signature) took an average of 777.1 milliseconds with 2938 bytes of data to create a new contract based on 10 experiments. The GDPR contract compliance verification took an average of 516.6 milliseconds based on 5 experiments. Similarly, in [9], a consent creation process took 7.3 s while compliance checking based on consent took 6.6 s. One of the main factors that affected the performance times is the complexity of contracts, which contain noticeably more information than consent.

## 8. Conclusions

In this paper, we presented the smashHitCore ontology that goes beyond existing work by representing both consent and contracts as legal GDPR bases for sensor data processing and sharing. The ontology is openly available and currently used as a schema for the smashHit KG (details in [72]). The utilisation of smashHitCore for consent and contract visualisation (see [61,62,73]) and for automated GDPR compliance (see [9,74]) has proven the ontology’s ability to successfully link two complex GDPR legal bases (consent and contracts) for our UC1 and UC2 in a meaningful way. The evaluation results have also shown that GDPR compliance verification can be performed in a reasonable amount of time with smashHitCore as the main underlying data model.

The following limitations of smashHitCore can be noted. Our ontology focuses primarily on GDPR-compliant consent- and contract-based vehicle sensor data sharing and processing in two use cases, while GDPR is the leading law on EU citizens’ data protection, each EU member state has its own country-specific legislations, procedures and data protection authorities. Finally, we have focused our work primarily on web-based (digital) consent and contracts thus processes such as their digitization have not been semantically modelled.

Ontologies as a type of digital asset require maintenance and updates over time. Future work might include expanding the scope of smashHitCore based on, but not only, its continuous use in UC1 and UC2. For example, extending the class *dpv:Purpose*, in smashHitCore, with different medical purposes and treatments from the Informed Consent Ontology (ICO) [75], which may or may not be covered by one’s insurance policy. Currently, smashHitCore reuses the class *dpv:Risk*. We plan to define specific risks when it comes to (personal) sensor data sharing in UC1 and UC2 as well. Extending the smashHitCore ontology with the DALICC [76,77] semantic model for digital licensing is also planned as future work. Licensing with DALICC can reduce transaction costs [77] and can improve data sharing of digital assets (e.g., contracts, personal datasets). An idea to be explored is the use of an upper ontology such as the Basic Formal Ontology (BFO) [78] for alignment of our ontology with others to support smashHitCore’s wider reuse. This can be conducted by aligning different processes that smashHitCore models (e.g., data processing, data sharing, giving and revoking consent, signing contracts, compliance verification) with BFO’s *Process* class at different levels. For instance, GDPR compliance verification is a complex process that can be divided into sub-processes such as identity verification, informed consent request, data use traceability etc. Each sub-process has temporal characteristics that are essential for performing adequate GDPR compliance verification. Although all this information can already be captured with smashHitCore, BFO’s classes *Process* and *ProcessProfile* can enhance the granularity of process details and simplify the reuse of smashHitCore beyond its current scope (BFO version 2.0 specification in [79]).

## Figures and Tables

**Figure 1 sensors-23-06188-f001:**
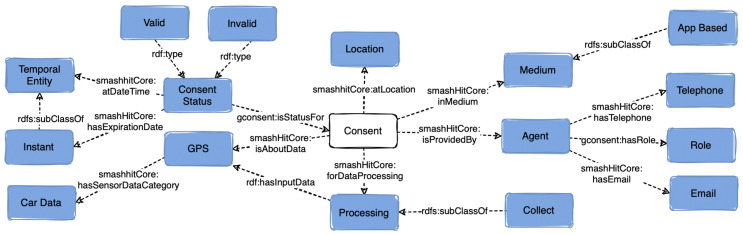
Overview of the class *Consent* and the classes related to it in smashHitCore.

**Figure 2 sensors-23-06188-f002:**
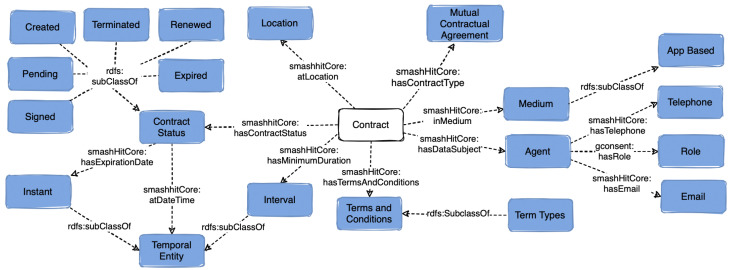
Overview of the class *Contract* and the classes related to it in smashHitCore.

**Figure 3 sensors-23-06188-f003:**
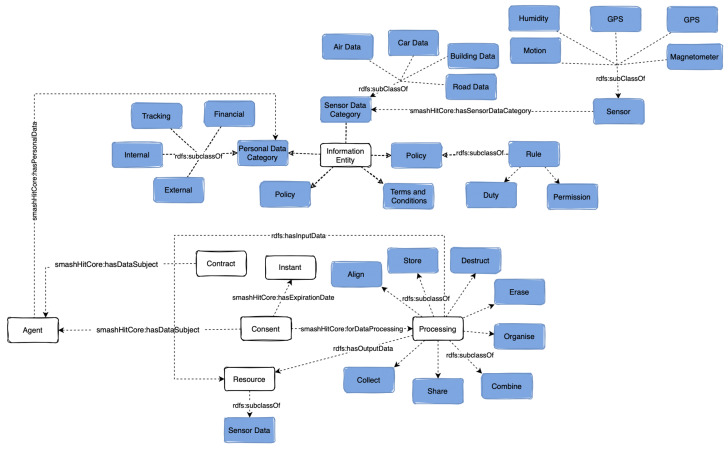
Class *Information Entity* and its connection to other concepts.

**Figure 4 sensors-23-06188-f004:**
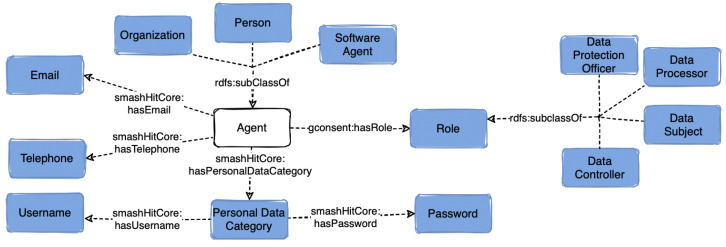
Overview of class *Agent* and the classes related to it in smashHitCore.

**Table 1 sensors-23-06188-t001:** Evaluation of smashHitCore with competency questions.

Questions	Relevant smashHitCore Concepts and Object Properties
**Informed consent questions:**	
What is the purpose of the consent?	*gconsent:Consent, dpv:Purpose, dpv:hasPurpose, smashHitCore:hasContext, smashHitCore:atLocation, smashHitCore:inMedium, smashHitCore:isAboutData*
What is the duration of the given consent?	*smashHitCore:consentID, smashHitCore:atDateTime, smashHitCore:hasExpirationDate, smashHitCore:hasRetentionDate*
What is the status of consent?	*smashHitCore:consentID, gconsent:StatusInvalidFofProcessing, gconsent:Withdrawn, smashHitCore:Granted, gconsent:isStatusFor, gconsent:hasStatus*
What is the context of consent?	*smashHitCore:consentID, smashHitCore:Context, smashHitCore:hasContext, smashHitCore:atLocation, smashHitCore:atDateTime, smashHitCore:inMedium*
How was consent requested, granted, revoked?	*smashHitCore:consentID, gconsent:Consent, gconsent:Medium, smashHitCore:inMedium, consent:AppBased, consent:Audio, consent:Video*
Who provides the consent?	*gconsent:Consent, smashHitCore:consentID, prov:Agent, smashHitCore:agentID dcat:Role, smashHitCore:DataSubject, smashHitCore:isProvidedBy*
**Data source and data processing questions:**	
What type of data is consent requested for?	*dpv:DataSource, dcat:Resource, smashHitCore:SensorDataCategory, smashitCore:hasContext, smashhitCore:hasMetadata*
What processing will be performed to the data?	*dpv:Processing, smashHitCore:hasMethod, dcat:Resource, dpv:Align, dpv:Erase, dpv:Analyse, dpv:Alter, dpv:Consult, dpv:Record, dpv:Restrict, rdf:hasInputData, rdf:hasOutputData*
What is the source of the data?	*dpv:DataSource, smashHirCore:Device, sensor:Sensor, smashHitCore:SensorDataCategory, smashHitCore:hasSensorDataCategory*
Who is responsible for the data processing?	*prov:Agent, prov:Organization, prov:Person, prov:SoftwareAgent, gconsent:hasRole, smashHitCore:DataController, smashHitCore:DataProvider*
**Agent questions:**	
What entities are involved with the data?	*prov:Agent, prov:Organization, prov:Person, prov:SoftwareAgent, rdf:Role, gconsent:hasRole, dcat:hadRole, smashHitCore:DataController, smashHitCore:agentID, smashHitCore:signatureID*
Who is the data controller?	*rdf:Role, gconsent:hasRole, dcat:hadRole, smashHitCore:DataController, smashHitCore:signatureID*
Who is the data subject?	*rdf:Role, smashHitCore:DataSubject, gconsent:hasRole, dcat:hadRole, smashHitCore:signatureID*
How to identify a person?	*dpv:PersonalDataCategory, dpv:Identifying, dpv:Name, dpv:OfficialID, dpv:UID, dpv:Username, dpv:Password, dpv:Contact, smashHitCore:hasContactPoint, dpv:EmailAddress, dcat:hadRole, gconsent:hasRole, smashHitCore:agentID, smashHitCore:signatureID*
How to identify an organisation?	*dpv:PersonalDataCategory, dpv:Identifying, dpv:Name, dpv:Contact, smashHitCore:hasContactPoint, dpv:EmailAddress, dcat:hadRole, gconsent:hasRole, smashHitCore:agentID*
**Contract questions:**	
What is the type of the contract between the entities?	*fibo-fnd-agr-agr:MutualContractualAgreement, smashHitCore:BusinessToBusiness, smashHitCore:BusinessToConsumer*
Who is involved in the contract?	*prov:Agent, prov:Organization, prov:Person, prov:SoftwareAgent, gconsent:hasRole, dcat:hadRole, dcat:Role, smashHitCore:DataController, smashHitCore:DataSubject, smashHitCore:agentID*
What is the duration of the contract?	*fibo-fnd-agr-smashHitCore:contractID, ctr:hasEffectiveDate, smashHitCore:hasCreationDate, smashHitCore:hasEndDate, smashHitCore:hasExpirationDate, smashHitCore:hasMinimumDuration*
What does a contract include?	*fibo-fnd-agr-smashHitCore:contractID, ctr:hasContractualElement, smashHitCore:TermsAndConditions, odrl:Policy, odrl:Duty, smashHitCore:Obligations, smashHitCore:obligationID, smashHitCore:hasSignatures, fibo-fnd-agr-ctr:hasBeneficiary*
What is the status of a contract?	*smashHitCore:contractID, smashHitCore:ContractStatus, smashHitCore:hasContractStatus, smashHitCore:Created, smashHitCore:Updated, smashHitCore:Expired, smashHitCore:Pending, smashHitCore:Renewed, smashHitCore:Signed, smashHitCore:Terminated*
What are the states of the contract obligations?	*smashHitCore:contractID, smashHitCore:Obligations, smashHitCore:obligationID, smashHitCore:ObligationState, smashHitCore:hasPendingState, smashHitCore:hasFulfilled, smashHitCore:hasInvalid, smashHitCore:hasValid*

## Data Availability

The smashHitCore ontology is available at https://smashhiteu.github.io.

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
