# Peer review of "The smashHitCore Ontology for GDPR-Compliant Sensor Data Sharing in Smart Cities"

_sensors, 2023, doi:10.3390/s23136188_

Round 1

Reviewer 1 Report

Good work, nicely presented and a pleasurable read. 

The results are somehow limited, since the future work will focus on expanding the scope of the ontology, it might have been better to publish with more results. 

Section 3 is very complete and thorough, the fact that it is based on a previous paper by the authors is not an issue but the references should be updated, it is lacking more recent work. 

1. What is the main question addressed by the research?

The main objective of the paper is to present an ontology for GDPR-compliant sensor data sharing in smart cities.

2. Do you consider the topic original or relevant in the field? Does it address a specific gap in the field? 

The topic is interesting and has a clear value. 

3. What does it add to the subject area compared with other published material? 

It builds on previous work on GDPR consent ontologies. The model includes two of six GDPR legal bases for data processing and future work will expand this scope.  

4. What specific improvements should the authors consider regarding the methodology? What further controls should be considered? 

The paper would be much stronger if it addressed the future work as well. As is, it is a small intermediary step.

5. Are the conclusions consistent with the evidence and arguments presented and do they address the main question posed? 

Yes, the conclusions are logical and relevant. 

6. Are the references appropriate?

The references could be improved, since the litterature study is based on previous work but references should be updated. 

7. Please include any additional comments on the tables and figures.

Author Response

Thank you for this review. Please see the attachment for details.

Reviewer 2 Report

The paper lacks information in section 6. about the evaluation or validation of the "smashHitCore" ontology, which hinders understanding of its performance, accuracy, scalability, and interoperability. More details on these aspects would enhance the ontology's effectiveness and reliability.

The abstract provides inadequate description of the use cases, UC1 and UC2, making it difficult to understand their relevance to the ontology. Including examples and context would help readers grasp the practical applications and benefits of the ontology.

The alignment process of the "smashHitCore" ontology with other ontologies, like BFO, is mentioned but lacks clarity. Elaborating on the alignment methodology and its implications would improve the paper's comprehensibility.

There is no detailed comparative analysis with existing ontologies or frameworks addressing similar concerns. Including a brief discussion of related works would highlight the novelty and advantages of the proposed ontology, providing readers with a broader perspective.

In summary, the paper emphasizes the "smashHitCore" ontology's relevance to GDPR and its potential for extensibility. However, it needs improvement in providing evaluation details, comprehensive use case descriptions, clarification of the alignment process, and inclusion of a comparative analysis.

Author Response

(The authors gave the same response as above.)

Reviewer 3 Report

The research article (current version) is needs to be reorganized and improved. After reviewing the article, it is still not clear about the novelty of this research. The authors should clarify for the ease of readers about the novelty, and main of this study where necessary. The research contribution(s), novelty should be highlighted throughout the paper and mainly in the introduction and conclusion.

 Some queries and suggestions are give as under:

What is the novel contribution in the article?

Abbreviation GDPR is used in title and multiple times in abstract but it has been declared later in the introduction.

The is no need to cite the existing work in the abstract as well as in conclusion.

Line 16...  (i.e. consent, contract, legal obligation, vital interest, public i.e can be repalted in including or any other better word ...

Line 94….  Both are described in detail in Section 2.2 and Section 2.1 respectively. Better to rewrite the sentence.

Parenthesis has been overused in the first paragraph of the introduction section.

Line 137 try to write clear sentences......... One of the earliest GDPR ontologies for consent is the Consent and Data Management Model (CDMM) by Fatema et al. [18].

Line 138-139... better to rewrite the sentence ......CDMM (available since 2017), represents consent as an entity within a privacy policy. The ontology is suitable to model consent through its life-cycle [10] (i.e. from request to withdrawal).

Line 161....... reusing the Semantic Sensor Network ontology (SSN) [24]. LloPY is not openly available.

Line 163….. The Business Process Re-engineering and Functional Toolkit for GDPR Compliance (BPR4GDPR) [25]. Try to follow a standard of declaring abbreviations.

Table 1 is cross-referenced on page no. 6 and its depicted-on page no. 11

Unnecessary line spacing on Page no. 11  

Line 168 ………. Consent is modelled as a policy and is used for privacy compliance checking by both SPECIAL’s Usage Policy Language (SPL) [26] and SPECIAL Policy Log Vocabulary (SPLog),

line 187...... Repetitively declared abbreviations and citations. .... Another widely used sensor ontology is the Semantic Sensor Network Ontology (SSN)[24] [24],

Line 238 Our survey in [10] and the related work overview in Section 3 have shown that there are many ontologies for consent as a GDPR basis for data processing. Better to rewrite the sentence.

Line 239 …GConsent focuses … unable to understand word GConsent.

What is the outcome of the conducted research is not highlighted in the conclusion.

Moderate English editing is required

Author Response

(The authors gave the same response as above.)

Round 2

Reviewer 3 Report

Authors are advised to avoid grammatical errors such as, line-45, ontologies help translate human knowledge into a machine

Authors are advised to avoid grammatical errors such as, line-45, ontologies help translate human knowledge into a machine

Author Response

Dear reviewers and editor(s),

The authors have proofread the manuscript to ensure its readability. The English language has been improved where needed. The latest updates for this manuscript's minor revision have been highlighted in green colour (please see the attachment).

Kind regards,

The authors
